# *Aspergillus* Enhances Eosinophil and Neutrophil Extracellular DNA Trap Formation in Chronic Rhinosinusitis

**DOI:** 10.3390/ijms242417264

**Published:** 2023-12-08

**Authors:** Seung-Heon Shin, Mi-Kyung Ye, Dong-Won Lee, Mi-Hyun Choi, Sang-Yen Geum

**Affiliations:** Department of Otolaryngology-Head and Neck Surgery, School of Medicine, Daegu Catholic University, Daegu 42472, Republic of Korea; miky@cu.ac.kr (M.-K.Y.); neck@cu.ac.kr (D.-W.L.); leonen@hanmail.net (M.-H.C.); saye60@naver.com (S.-Y.G.)

**Keywords:** *Aspergillus fumigatus*, chronic rhinosinusitis, eosinophil, neutrophil, extracellular trap, reactive oxygen species

## Abstract

Chronic rhinosinusitis (CRS) is characterized by inflammatory cell infiltration in the sinonasal mucosa. Eosinophil and neutrophil extracellular traps (EETs and NETs, respectively) are prominently found in CRS. This study aimed to investigate the effect of airborne fungi, *Alternaria alternata* and *Aspergillus fumigatus*, on EET and NET formation. Nasal epithelial cells, eosinophils, and neutrophils were isolated from eosinophilic CRS (ECRS), non-ECRS (NECRS), and healthy control. We determined eosinophil and neutrophil transepithelial migration after fungal treatment. We then determined the release of EETs and NETs by fungi using Sytox Green staining and determined the role of reactive oxygen species (ROS) using ROS inhibitors. We identified more abundant EETs and NETs in ECRS than in NECRS. *A. alternata* and *A. fumigatus* enhanced eosinophil and neutrophil transepithelial migration. *A. fumigatus* strongly induced EET and NET formation in CRS and, simultaneously, suppressed fungal metabolic activity. EET formation in CRS is associated with nicotinamide adenine dinucleotide phosphate (NADPH)–oxidase and NET formation with NADPH–oxidase and mitochondrial ROS. *A. fumigatus*, but not *A. alternata*, induced EET and NET formation, and peripheral blood eosinophils and neutrophils exhibited different immune responses against *A. fumigatus* following the inflammatory status of the host. *Aspergillus-fumigatus*-induced EET and NET formation plays a crucial role in CRS pathogenesis.

## 1. Introduction

Extracellular traps (ETs) serve as a key component of the innate immune response against external pathogens. They consist of dense DNA, histones, granule proteins, and several inflammatory mediators [1]. Not only neutrophils but also other cells, including eosinophils and mast cells, produce ETs [2,3]. Neutrophils and eosinophils are both granulocytes and are known to act as effector cells in the immune response against infections and pathological conditions. Neutrophil ETs (NETs) and eosinophil ETs (EETs) are beneficial for host defense owing to their ability to eliminate pathogens, while the released nuclear-derived DNA and granular enzymes can have cytotoxic effects on airway epithelial cells. This can increase secretion viscoelasticity and promote local inflammation and tissue damage [4]. The characteristic features of type 2 chronic rhinosinusitis (CRS) include the presence of tissue eosinophilia, neutrophilia, and increased NETs and EETs [5]. EETs in CRS are closely related to disease severity and CRS prognosis. NETs are found in the subepithelial layer of sinonasal mucosa and are associated with refractoriness in non-eosinophilic CRS (NECRS) [5,6]. NETs and EETs play a crucial role in the development of CRS or CRS with nasal polyps (CRSwNP) and can be targeted for therapeutic interventions.

Neutrophils and eosinophils act as an effector cell in the removal of fungal pathogens. Neutrophils eliminate fungi through degranulation, phagocytosis, reactive oxygen species (ROS) generation, cytokine production, and NET formation [7,8]. Fungi induce NET formation through nicotinamide adenine dinucleotide phosphate (NADPH)–oxidase and the ROS-dependent response or through NADPH–oxidase-independent opsonization by human serum for non-phagocytic effector mechanisms. NETs demonstrate fungicidal properties against many pathogenic fungi as well as the fungistatic effect against *Aspergillus*, thereby preventing their further spread [9,10,11]. Fungi induce eosinophil degranulation through interaction with β-glucan and β2-integrin or fungal protease with protease-activated receptors [12,13]. EETs can develop in both ROS-dependent and ROS-independent manner. The response promotes epithelial cell-derived cytokine production, such as thymic stromal lymphopoietin (TSLP), which induces the release of EETs in a NADPH–oxidase-dependent manner [14]. *Aspergillus* induces the release of EETs in an NADPH–oxidase-independent manner [15].

Airborne fungi are ubiquitous and prominently associated with airway diseases. Among various pathogenic fungi, *Aspergillus* and *Alternaria* are frequently isolated from airway secretions, and they interact with airway mucosa and inflammatory cells [16]. Patients with CRS tend to have a higher fungal burden and exhibit abnormal or inappropriate immune responses to fungi [17]. Clearance of inhaled fungal conidia in the airway mucosa by neutrophils or eosinophils is a crucial innate defense mechanism. Neutrophils and eosinophils play a crucial role in antifungal immunity and immunomodulatory function in fungal-associated inflammatory diseases, and they release ETs in response to fungal pathogens [11]. This study aims to investigate the effects of *Aspergillus* or *Alternaria* on neutrophils and eosinophils migration as well as the release of ETs by these cells in response to fungi.

## 2. Results

### 2.1. EET and NET Expression in Patients with CRSwNP

Eosinophilic CRS (ECRS) is characterized by blood and tissue eosinophilia and highly viscous nasal secretions with early olfactory dysfunction. This study detected eosinophils using galectin 10 staining, and neutrophils were stained with elastase. Although nuclear DNA traps were abundant in nasal secretions, we can also detect EETs and NETs in sinus mucosa in ECRS. The fluorescence intensity of NETs was 1.4 times higher, and that of EET was 1.7 times higher in patients with ECRS compared to patients with NECRS. Additionally, cells positive for EETs and NETs were also greater in patients with ECRS than in patients with NECRS (26.8 ± 4.9, 6.5 ± 1.7, respectively, *p* = 0.007 for EET and 33.2 ± 8.2, 10.6 ± 2.7, respectively, *p* = 0.034 for NET) (Figure 1).

### 2.2. Fungal-Induced Eosinophil and Neutrophil Migration

Fungal protease can affect the expression of tight junction molecules in nasal epithelial cells [18]. Eosinophils and neutrophils were placed in the basolateral portion after 2 h treatment of nasal epithelial cells with fungal conidia to induce migration for 4 h. Nasal epithelial cell stimulation with *Aspergillus fumigatus* and *Alternaria alternata* resulted in an increased eosinophil and neutrophil migration to the apical side. Both *Alternaria* and *Aspergillus* notably amplified eosinophil migration in epithelial cells derived from patients with ECRS and those derived from the inferior turbinate. However, the presence of fungi did not exert any discernible influence on eosinophil migration in epithelial cells from NECRS (Figure 2A). Both *Alternaria* and *Aspergillus* significantly enhanced neutrophil migration in epithelial cells from patients with ECRS and patients with NECRS, as well as in cells derived from the inferior turbinate (Figure 2B). Notably, *Aspergillus* caused a substantially greater increase in the migration of both eosinophils and neutrophils compared to *Alternaria* when comparing epithelial cells from the inferior turbinate.

Fungal protease is known to induce not only chemical mediator production but also directly influence epithelial barrier functions [18,19]. Fungal protease-induced ROS production is associated with epithelial barrier dysfunction. The protease activity of *Alternaria* and *Aspergillus* (10^6^ conidia) was similar to that of 2.0 μg/mL of trypsin (2.3 μg/mL for *Alternaria* and 1.9 μg/mL for *Aspergillus*, Figure 3A). TER can represent epithelial cells permeability and molecular structure. After 4 h incubation of nasal epithelial cells, TER was increased by approximately 21.8% compared to time zero. TER decreased to 69.5% and 68.0% compared to baseline TER at 0 h when the nasal epithelial cells were treated with *Alternaria* or *Aspergillus*. TER returned to the non-treated level when the epithelial cells were pretreated with glutathione, a ROS scavenger, or heat-inactivated fungi (Figure 3B).

### 2.3. Fungal-Induced EET and NET Formation

CRS, especially ECRS, exhibited a significant increase in both EETs and NETs in sinonasal mucosa. We investigated whether airborne fungi could influence EET and NET development in CRS. Peripheral blood eosinophils and neutrophils were isolated from patients with ECRS, those with NECRS, and normal healthy volunteers. They were incubated with *A. fumigatus* or *A. alternata* at a 1:10 ratio for 6 h, then ET formation was determined using a fluorometric method. EETs were significantly increased as a result of *Aspergillus* in all three groups. However, *Alternaria* did not induce EET formation from peripheral blood eosinophils. The amount of EETs was twice as high in NECRS and thrice as high in ECRS compared to in the non-treated group (Figure 4A,C). *Aspergillus* strongly induced NETs from peripheral blood neutrophils in all three groups. However, *Alternaria* only induced NETs in NECR. The number of *Aspergillus*-induced NETs was approximately five times higher in NECRS and six times higher in ECRS. The number of *Alternaria*-induced NETs in NECRS was twice as high as in the non-treated group (Figure 4B,D).

ETs are associated with the release of granule proteins and chemical mediators. We attempted to measure eosinophil cationic protein (ECP) and neutrophil elastase using the ELISA method, but the protein levels were too low for comparison (0.98–1.77 ng/mL for ECP and 0.48–1.03 ng/mL for neutrophil elastase). We also determined interleukin (IL)-8 protein levels in culture supernatants. IL-8 levels experienced a substantial increase as a result of stimulation with *Alternaria* and *Aspergillus* in all three groups in EETs (Figure 5A). *Aspergillus* induced a larger amount of IL-8 production from eosinophils in ECRS and NECRS than *Alternaria*. IL-8 levels also experienced a substantial increase as a result of stimulation with *Alternaria* and *Aspergillus* in all three groups in NETs (Figure 5B). Additionally, *Aspergillus* induced a larger amount of IL-8 production from neutrophils in ECRS and NECRS than *Alternaria*.

### 2.4. Effects of ROS Inhibitors on EET and NET Formation

Eosinophils or neutrophils isolated from ECRS patients were pretreated with DPI, a nonselective inhibitor for the production of ROS, and/or MitoTempo, an inhibitor of mitochondrial ROS, to determine the role of ROS in EET development. First, we measured intracellular ROS levels in eosinophils and neutrophils. Cells were pretreated with 30 μM of DPI for 30 min and then stimulated with fungi for 6 h or PMA for 2 h. ROS levels were measured using dihydrorhodamine. Intracellular ROS in eosinophils and neutrophils tended to increase in a time-dependent manner as a result of fungi and PMA. DPI tended to suppress the ROS production in eosinophils and neutrophils (Figure 6A,B). Although ROS production from eosinophils or neutrophils in NECRS patients was not as strong as that in ECRS patients, it exhibited similar patterns between the two patient groups (Appendix A).

EET and NET formation was significantly higher, as determined by Sytox Green staining, after 6 h of eosinophil or neutrophil treatment with *Aspergillus*. *Aspergillus*-induced EETs were significantly inhibited by the NADPH–oxidase inhibitor DPI, but a mitochondrial ROS inhibitor, MitoTempo, did not influence EET development. Both DPI and MitoTempo significantly inhibited *Aspergillus*-induced NET formation. *Alternaria* did not influence EET and NET formation. Both DPI and MitoTempo significantly inhibited PMA-induced EET and NET formation (Figure 6C,D). The effect of ROS inhibitors on EET and NET formation showed a similar pattern in cells isolated from NECRS and ECRS patients (Appendix A).

## 3. Discussion

Eosinophils and neutrophils have a similar biological activity, acting as effector cells against fungal pathogens and modulating adaptive immune responses through extracellular trap formation [11]. *Alternaria* and *Aspergillus* are commonly found in the nasal secretion of both healthy volunteers and patients with CRS [20]; thus, this study elucidated the role of these airborne fungi in EET and NET formation in CRS. Sinonasal tissue from patients with ECRS exhibited a higher number of NETs and EETs compared to NECRS. *Aspergillus*, but not *Alternaria*, induced EET and NET formation, which were more pronounced in patients with ECRS compared to those with NECRS and to healthy volunteers.

Airborne fungi enter the airways through inhalation, and most of them are removed by the innate mucosal defense system, including mucociliary clearance, fungicidal enzymes, and innate immune cells. Airway epithelial cells induce the production of chemical mediators and facilitate pathogen access to the target tissue or the migration of inflammatory cells into the airway lumen in pathological conditions with increased fungal contact [19,21]. Fungal cell wall components, especially protease, can induce epithelial barrier dysfunction with ROS overproduction [18]. The protease activity of 10^6^ conidia of *Alternaria* and *Aspergillus* was similar to that of 2.0 μg/mL of trypsin. Epithelial TER decreased after 4 h of fungal conidia treatment. TER experienced a more substantial decrease in ECRS patients compared to the other two groups (30–35%, 17–25%, and 15–20% in the ECRS, NECRS, and control groups, respectively), although failing to attain statistical significance. *Alternaria* and *Aspergillus* enhanced both eosinophil and neutrophil migration in the ECRS and control groups. However, only the neutrophil migration increased as a result of fungi in the NECRS group. These discrepancies may be associated with the strength of molecular interactions within the apical junctional complex or with the difference in the effect of fungi on eosinophil-associated chemokine production from nasal epithelial cells. Furthermore, eosinophil migration was more strongly induced by fungi in the ECRS group. We cannot explain the reasons for these differences, but the pronounced eosinophil migration induced by fungi is thought to result from barrier dysfunction, which may involve a molecular or structural change of tight or apical epithelial cell junctions due to fungal protease or other components, as well as the abundant production of chemical mediators from epithelial cells compared to other conditions. *Aspergillus* strongly induced neutrophils migration compared to *Alternaria*. This phenomenon may be attributable to the strong induction of neutrophilic innate immune responses by *Aspergillus* with respect to the identification and elimination of fungal organisms, especially in normal nasal epithelial cells. Increased intracellular ROS is generally associated with nasal epithelial barrier dysfunction [18]. ROS production was inhibited by treatment with the ROS scavenger glutathione, indicating that intracellular ROS is a contributing factor to epithelial barrier dysfunction and inflammatory cell migration in the nasal mucosa.

Eosinophils modulate the immune response in various infectious diseases and act as effector cells to protect the host against non-phagocytizable pathogens, such as parasites and fungi [22]. Eosinophils can remove fungi through both physical contact with β-glucan components of fungal cell walls or without physical contact through pattern-recognition receptors (PRRs) associated with eosinophil activation [12,23]. PRRs, particularly the β2-integrin Mac-1, respond to fungi and mediate fungal-induced EET release [15]. *Alternaria* can induce activation and release of toxic granule protein from eosinophils through physical interactions between CD11b and the β-glucan of fungi [12]. However, this study revealed that *Alternaria* could not influence EET development, while *Aspergillus* strongly induced EETs in vitro. Fungi exhibit remarkable diversity in terms of structure and components, and they interact with the host and immune cells in various ways. While *Alternaria* can activate eosinophils through various way, it lacks the ability to actively participate in the development of ETs as observed in *Aspergillus*. *Aspergillus*-induced EET formation was much more pronounced in ECRS. This indicates that, in ECRS, eosinophils may be more prone to respond strongly and quickly to *Aspergillus*. In this study, approximately half of ECRS patients had asthma. Although, we did not directly compare the impact of asthma comorbidity on EET formation, the biological properties of eosinophils in asthmatic patients may predispose to the development of EET. The ROS inhibitor DPI treatment, but not the mitochondrial ROS inhibitor MitoTempo, strongly inhibited *Aspergillus*-induced EET formation. This result differs from the previous study which indicated that *A. fumigatus* induces the release of EETs in an NADPH–oxidase- and mitochondrial-ROS-independent manner [15]. This discrepancy may be due to the use of eosinophils isolated from patients with ECRS, whose biological condition may be different from those of healthy volunteers.

Neutrophils are essential for the clearing of fungi through the phagocytosis of conidia, degranulation, and oxidative burst [8,10]. NET formation is involved in fungal killing and entrapment [11]. The intensity of *Aspergillus-fumigatus*-induced NET formation was highest in ECRS, but *A. fumigatus* also induced NET formation in patients with NECRS and in heathy controls. *Alternaria* only induced NET formation in patients with NECRS. NETs were found to be abundant in sinonasal tissue with fewer EETs in NECRS, and they may play a critical role as part of an innate immune response against airborne fungi, such as killing or inhibiting fungal growth and germination. NADPH–oxidase activation plays a crucial role in NET formation and in the effector response to external pathogens [24]. This study revealed that both NADPH–oxidase and mitochondrial ROS inhibitors significantly inhibited *Aspergillus-fumigatus*-induced NET formation.

EET and NET accompany rapid cell death characterized by cell membrane rupture and the release of intracellular granule proteins and chemical mediators. We measured IL-8 and ECP for eosinophils and elastase for neutrophils. However, due to the low sensitivity of commercially available ELISA kits for ECP and elastase in the ng/mL range, measuring and comparing them statistically was challenging. *Aspergillus* strongly induced IL-8 production compared to *Alternaria* both in EETs and NETs. Because *Aspergillus* strongly enhance the development of EET and NET compared to *Alternaria*, *Aspergillus* strongly induced the release of a larger amount of intracellular IL-8 from eosinophils and neutrophils. IL-8 levels were significantly higher in eosinophils or neutrophils from patients with CRS than in those from healthy volunteers, and IL-8 levels within CRS were much higher in ECRS patients than in NECRS patients. The IL-6 and tumor necrosis factor (TNF)-α levels in EETs were also higher in ECRS patients than in NECRS patients (Appendix A). Eosinophils and neutrophils that migrated to the fungi may have been in a preactivated state or prone to producing larger amounts of intracellular contents in patients with CRS. Eosinophils and neutrophils might be more strongly activated by fungi in ECRS patients than in NECRS patients. ET formation by eosinophils and neutrophils is known to contribute to pathogenic fungi removal and may influence fungal metabolic activity. The IL-8 study revealed that *Aspergillus* only induced a small amount of IL-8 production from eosinophils or neutrophils derived from healthy volunteers compared to patients with CRS. This indicates that *Aspergillus* could not strongly induce the release of granule protein and chemical mediators compared to patients with CRS, which have fungicidal or fungistatic properties.

## 4. Materials and Methods

### 4.1. EET and NET Detection in CRSwNP

This study collected sinonasal polyp tissues from 30 patients with CRSwNP during endoscopic sinus surgery. We excluded patients who were younger than 18 years of age, had active inflammation, and had used antibiotics, antihistamines, or other medications for at least 4 weeks prior to the operation. The Institutional Review Board of Daegu Catholic Medical Center approved this study, and all the patients signed a consent form that outlined the study objectives. We classified ECRS as eosinophilic when more than an average of 70 eosinophils were found in the three densest areas on high-power field (×400) images. Table 1 shows the clinical and demographic characteristics of patients with CRSwNP. The average counts of EETs and NETs were analyzed in three representative fields characterized by the highest degree of cell infiltration throughout the sinus mucosa.

Immunofluorescence staining was used to detect EETs and NETs in CRSwNP. The deparaffinized tissue slides were treated with 0.3% Triton-X100 for 15 min and blocked with 5% bovine serum albumin (Merck, Darmstadt, Germany). The slides were incubated with histone H3 (citrulline R17) antibody (Abcam, Cambridge, UK) and neutrophil elastase (Santacruz, Dallas, TX, USA) to identify NET. The slides were incubated with histone H3 antibody and galactin-10 (Santacruz) for EET. After being washed with phosphate buffer saline (PBS), the section slides were incubated with Alexa Fluor^®^ 488-conjugated IgG (Invitrogen, Carlsbad, CA, USA) and cy3-conjugated IgE (Invitrogen). The cell nucleus was stained with Hoechst 33342 (Invitrogen). The Zeiss LSM 900 confocal microscope (Zeiss, Oberkochen, Germany) was used to determine image capture and analyses.

### 4.2. Isolation of Fungal Conidia

*Aspergillus fumigatus* (ATCC 46645) and *Alternaria alternata* (KCTC 26781) conidia were inoculated on potato dextrose/corn meal agar plates with cycloheximide for 5–7 days at 25 °C. Conidia were collected by scrapping the plate with sterile PBS that contained 0.05% tween-20 (Bio-Rad, Hercules, CA, USA). Eluates were centrifuged at 1000 rpm for 10 min, and pellet suspensions were filtered through a 40 μm cell strainer. Conidia suspensions at 2 × 10^7^/mL were left to dry at 45 °C, after which they were stored at −80 °C until required.

### 4.3. Isolation of Eosinophil and Neutrophil

Cells were isolated from heparinized venous blood from patients with ECRS, NECRS, and also from normal healthy volunteers. The Institutional Review Board of Daegu Catholic Medical Center approved this study, and all subjects signed a consent form outlining the study objectives. Neutrophils were isolated using the density gradient centrifugation method with 1.077 g/mL of percoll (Sigma-Aldrich, St. Louis, MO, USA). After collecting granulocyte and red blood cells (RBCs), the RBCs were removed using hypotonic water lysis. The purity and viability of neutrophils determined using trypan blue staining was >95%.

Negative immunomagnetic bead selection was used to isolate eosinophils. Heparinized blood was layered over 1.085 g/mL of percoll (Sigma-Aldrich). After RBC lysis, the granulocytes were mixed with anti-CD16 antibody conjugated with magnetic particles (Miltenyl Biotechnology, Sunnyvale, CA, USA) and passed through a magnetic field (Variomax, Miltenyl Biotechnology). The purity of the eosinophils determined using Randolph staining was >95%.

### 4.4. Primary Nasal Epithelial Cell Inverted Air-Liquid Interface (ALI) Culture

Primary nasal epithelial cells were isolated from the NP of subjects with ECRS (six men and two women, aged 51.3 ± 16.2 years), NECRS (five men and two women, aged 48.3 ± 13.5 years), and inferior turbinates of septal deviation (five men and three women, aged 47.6 ± 12.7 years). The Institutional Review Board of Daegu Catholic Medical Center approved this study, and all subjects signed a consent form that outlined the study objectives.

The specimens were placed in Roswell Park Memorial Institute (RPMI) 1640 medium supplemented with antibiotic–antimycotic (100×). Nasal mucosa was rinsed with RPMI 1640 supplemented with antibiotics and incubated with 0.1% protease (Sigma-Aldrich) for 16 h at 4 °C. The epithelial cells were isolated by gentle agitation, and cell suspensions were filtered through a No. 60 mesh cell dissociation sieve. The cells were suspended in defined keratinocyte serum-free medium (ThermoFisher Scientific, Waltham, MA, USA) supplemented with antibiotics. Cell suspensions (10^6^ cells/mL) were plated in six-well culture plates and placed in a 5% CO_2_ humidified incubator at 37 °C. The culture medium was changed after 24 h and every 2 days thereafter. The cells were seeded on 3.0 μm 12 mm transwell inserts (Corning Costar, Cambridge, MA, USA) when the epithelial cells cultures reached 80–90% confluence to make ALI culture model at a density of 1 × 10^5^ cells/well mixed with the 0.5% GrowDex^®^ (UPM Biochemicals, Helsinki, Finland) in a medium. The medium was retrieved to adapt the inverted ALI culture in the PneumaCult ALI basal medium (Stemcell, Vancouver, BC, Canada) once the epithelial cells grew to complete confluence. The transwell plates were positioned upside down to place the epithelial cells at the apical side. After the cell was seeded, it was incubated overnight in a medium, and the transwells were flipped back to the normal position in 24-well plates. PneumaCult-ALI Maintenance medium was added to an apical camber and PneumaCult Ex Basal Medium to the basal chamber of the wells when cell cultures reached 80% confluence. The medium was changed every second day after 14 days in ALI culture.

### 4.5. Transepithelial Migration of Eosinophil and Neutrophil

ALI-cultured apical nasal epithelial cells were treated with 2 × 10^6^ conidia/cm_2_ for 2 h and then excess conidia were washed out to determine the effect of *A. fumigatus* on cell migration. ALI-cultured cells were positioned upside down to place the apical nasal epithelial cells on the underside, and one hundred microliters of the eosinophils or neutrophils (5 × 10^6^/mL) were added at the basolateral side to enable the migration. After 4 h, apically migrated cells were counted by using light microscopy. To investigate the effect of eosinophil and neutrophil migration in the epithelial cells of the ECRS, NECRS, and normal control groups, cells isolated from the peripheral blood of each ECRS, NECRS, and normal control group were utilized.

The capacity of the fungi to disrupt epithelial barrier function was determined by measuring transepithelial resistance (TER) using EVOM-II (World Precision Instruments, New Haven, CT, USA). The values for cell-covered filters were calculated in standard units of ohms × square centimeter (Ω × cm_2_) after subtracting the resistance of blank filters, and TER was expressed as a percentage compared with the non-stimulated condition at time zero. The effect of ROS on the migration was determined by epithelial pretreatment with 5 mmol/L of glutathione.

### 4.6. Confocal Microscopic Evaluation of EETs and NETs

Purified eosinophils or neutrophils were cultured with *A. fumigatus* or *A. alternata* at a cell-to-fungi conidia ratio of 1:10 for 6 h at 37 °C. ETs were detected using Sytox Green (Life Technologies, Gaithersburg, MD, USA) staining in a FLUOstar Optima (BMG Labtech, Ortenberg, Germany) with a wavelength combination of excitation at 485 nm and emission at 538 nm. Cells, as a positive control, were treated with 50 nM of phorbol 12-myristate 13-acetate (PMA, Sigma-Aldrich) for 2 h. Cells were pretreated with 30 μM of diphenyleneiodonium (DPI, Sigma-Aldrich) or 500 μM of MitoTempo (Invitrogen, Carlsbad, CA, USA) as a ROS inhibitor to determine the role of ROS in ET development.

### 4.7. Quantification of ECP, Neutrophil Elastase, and IL-8

Cell-free supernatants were collected after a 6 h incubation with eosinophils or neutrophils and *A. fumigatus* or *A. alternata* or 2 h treatment with PMA at a cell-to-fungi conidia ratio of 1:10. ECP (MyBioSource, San Diego, CA, USA) and IL-8 (Invitrogen) were measured for EETs using a commercially available enzyme-linked immunosorbent assay (ELISA) kit. Neutrophil elastase (Invitrogen) and IL-8 for NETs were measured following the manufacturer’s directions. The detection limits were 0.39 ng/mL for ECP, 0.16 ng/mL for neutrophil elastase, and 2 pg/mL for IL-8.

### 4.8. Quantification of ROS in Eosinophils and Neutrophils

This study collected sinonasal polyp tissues from 30 patients with CRSwNP during endoscopic sinus surgery. We excluded patients who were younger than 18 years of age, had active inflammation, and had used antibiotics, antihistamines, or other medications for at least 4 weeks prior to surgery. The Institutional Review Board of Daegu Catholic Medical Center approved this study, and all the patients signed a consent form that outlined the study objectives. We classified ECRS as eosinophilic when more than an average of 70 eosinophils were found in the three densest areas on high-power field (×400) images. Table 1 shows the clinical and demographic characteristics of patients with CRSwNP.

### 4.9. Determination of Fungal Protease

The protease activity of *A. fumigatus* and *A. alternata* was determined using a protease activity assay kit (Cayman, Ann Arbor, MI, USA). Fungal conidia were placed in 96-well black plates with a protease substrate for 20 min at RT. A total of 2.0 μg/mL trypsin was used as a standard sample. The value of protease activity was determined with an excitation wavelength of 485 nm and an emission wavelength of 520 nm using FLUOstar Optima.

### 4.10. Statistical Analysis

The Statistical Package for the Social Sciences version 25.0 (IBM Corp., Armonk, NY, USA) was used to analyze data obtained in the experiments, with data presented as the mean ± standard deviation. The clinical profiles of patients with CRS were compared using the Mann–Whitney U test and the two-sample *t*-test. Cell migration, EET and NET formation, IL-8, and fungal metabolic activity were analyzed using a one-way analysis of variance, followed by a Dunnett’s test. Differences with a *p*-value of ≤0.05 were considered statistically significant.

## 5. Conclusions

Chronic and repeated fungal exposure in CRS may induce eosinophil and neutrophil recruitment to the nasal mucosa and the development of ET formation as part of the innate defense system against fungi. Tissue EET and NET counts were significantly higher in ECRS patients than in NECRS patients. We induced EET and NET formation using human blood eosinophils and neutrophils from patients with ECRS, those with NECRS, and normal healthy volunteers in response to *A. alternata* and *A. fumigatus* exposure. *Aspergillus* influenced the development of EET and NET formation, and ET formation was strongly induced in ECRS patients. Eosinophils and neutrophils suppressed *A. fumigatus* metabolic activity in CRS, which may be associated with the fungicidal activity of EETs and NETs. EET formation is associated with intracellular NADHP–oxidase, and NET formation is associated with both NADPH–oxidase and mitochondrial ROS. This study revealed that ECRS shows higher EET and NET formation, and peripheral blood eosinophils and neutrophils exhibit different immune responses against *A. fumigatus* according to the inflammatory status of the host. Identifying the mechanism of *Aspergillus-fumigatus*-induced EETs and NETs could provide important insights into CRS pathogenesis.

## Figures and Tables

**Figure 1 ijms-24-17264-f001:**
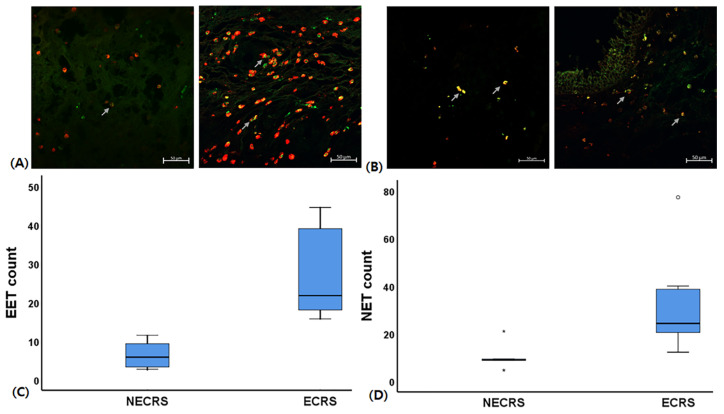
Eosinophil extracellular trap (EET) and neutrophil extracellular trap (NET) detection in patients with eosinophilic and non-eosinophilic chronic rhinosinusitis (ECRS and NECRS, *n* = 16 and 14, respectively). Representative images of the EETs (**A**) and NETs (**B**) in patients with NECRS and ECRS (20× magnification, scale bars = 50 μm). EETs were visualized using histone H3 (green) and galatin-10 staining (red), while NETs were visualized using histone H3 and elastase staining (red). Arrows indicate cells that are positive for NETs and EETs. EET-positive (**C**) and NET-positive cells (**D**) were notably higher in patients with ECRS compared to those with NECRS (×200). Asterisks and dot represent Out-liers from our data.

**Figure 2 ijms-24-17264-f002:**
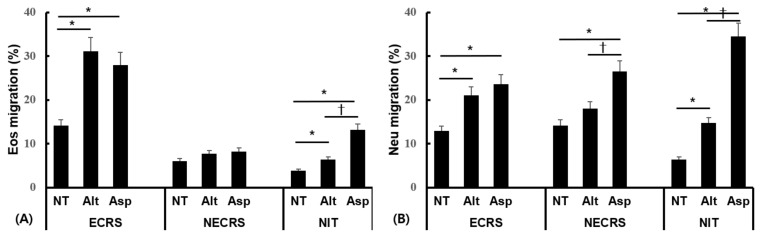
Effects of nasal epithelial cell exposure to *Alternaria alternata* (Alt) and *Aspergillus fumigatus* (Asp) on eosinophils (Eos) (**A**) and neutrophils (Neu) (**B**) migration. Nasal epithelial cells were isolated from eosinophilic chronic rhinosinusitis (ECR), non-eosinophilic chronic rhinosinusitis (NECRS), and normal inferior turbinate (NIT). *Alternaria* and *Aspergillus* enhanced eosinophil and neutrophil migration through nasal epithelial cells, except for eosinophils in NECRS. NT: not treated, * *p* < 0.05 compared with NT, † *p* < 0.05 compared with Alt, *n* = 6.

**Figure 3 ijms-24-17264-f003:**
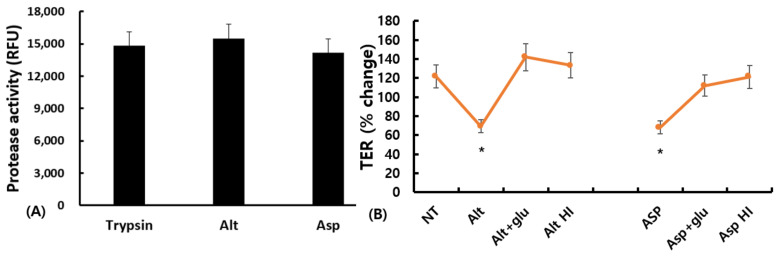
Protease activity of *Alternaria alternata* (Alt) and *Aspergillus fumigatus* (Asp) and their effect on transepithelial resistance (TER). Fungal protease activity was similar to that of 2.0 μg/mL of trypsin (**A**). TER decreased upon fungal treatment. TER returned to the baseline level when the nasal epithelial cells were treated with heat-inactivated (HI) fungi or pretreated with glutathione (glu) (**B**). NT: not treated, * *p* < 0.05 compared with NT, *n* = 6.

**Figure 4 ijms-24-17264-f004:**
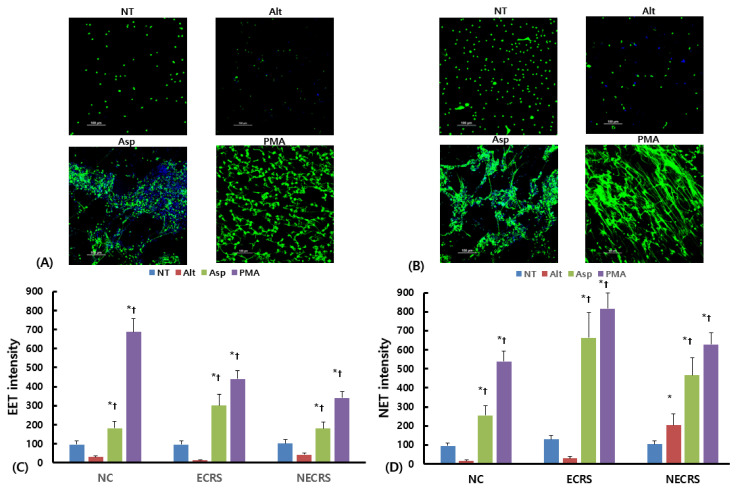
*Alternaria alternata* (Alt) and *Aspergillus fumigatus* (Asp) induced eosinophilic extracelluar trap (EET) and neutophilic extracellular trap (NET) formation in cells isolated from patients with eosinophilic chronic rhinosinus (ECRS), those with non-eosinophilic chronic rhinosinusitis (NECRS), and normal healthy volunteers (NC). (**A**,**B**) show the representative images of the EET and NET (40× magnification, scale bars = 100 μm). (**C**,**D**) show the intensity for EET and NET formation. *Aspergillus* strongly enhanced EET and NET formation. PMA was used as a positive control. NT: not-treated, * *p* < 0.05 compared with NT, † *p* < 0.05 compared with Alt, *n* = 5.

**Figure 5 ijms-24-17264-f005:**
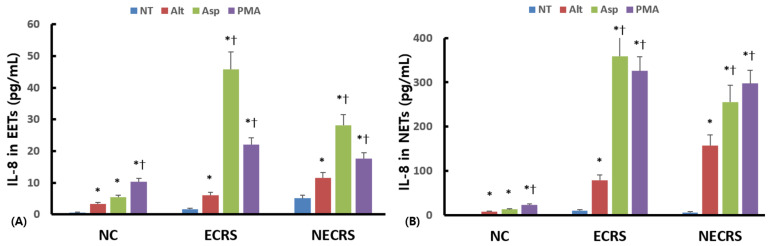
The concentration of interleukine-8 (IL-8) in culture supernatant after eosinophilic extracellular traps (EETs) (**A**) and neutrophilic extracellular traps (NETs) (**B**) formation by fungi. *Alternaria alternata* (Alt) and *Aspergillus fumigatus* (Asp) strongly induced IL-8 production from eosinophils and neutrophils in eosinophilic chronic rhinosinusitis (ECR) and non-eosinophilic chronic rhinosinusitis (NECRS). *Aspergillus* significantly enhanced IL-8 production compared to *Alternaria*. PMA was used as a positive control. NC: normal control, NT: not treated, * *p* < 0.05 compared with NC, † *p* < 0.05 compared with Alt, *n* = 5.

**Figure 6 ijms-24-17264-f006:**
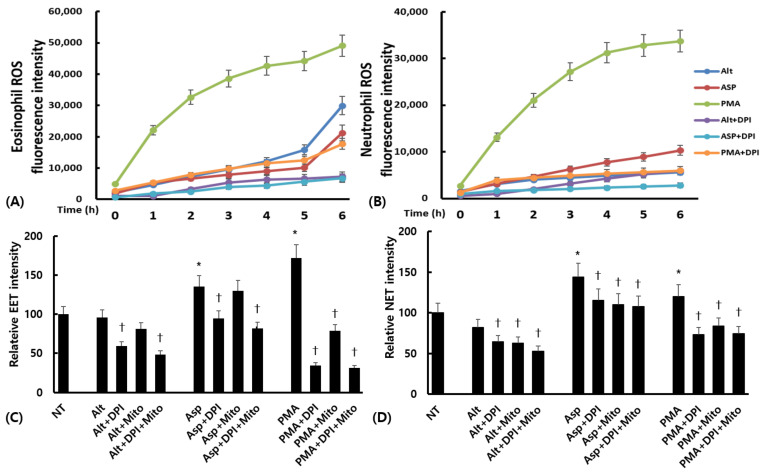
The effect of reactive oxygen species (ROS) inhibitors on EET or NET formation in eosinophilic chronic rhinosinusitis as a result of fungi. *Alternaria alternata* (Alt) and *Aspergillus fumigatus* (Asp) enhanced intracellular ROS production in a time-dependent manner (**A**,**B**). Nicotinamide adenine dinucleotide phosphate (NADPH)–oxidase inhibitor, i.e., diphenyleneiodonium (DPI), significantly suppressed both *Aspergillus*-induced EET and NET formation. Additionally, the mitochondrial ROS inhibitor, MitoTempo (Mito), only inhibited *Aspergillus*-induced NETs (**C**,**D**). PMA was used as a positive control. NT: not treated, * *p* < 0.05 compared with NT, † *p* < 0.05 compared with Alt, *n* = 5.

**Table 1 ijms-24-17264-t001:** Demographic and clinical profiles of chronic rhinosinusitis (CRS) in patients with nasal polyp.

Variables	ECRS (*n* = 16)	NECRS (*n* = 14)	*p*-Value
Sex			0.069
Male	12	10	
Female	4	4	
Age, year	53.8 ± 7.6	50.5 ± 13.0	0.403
CT finding			
Lund-Mackay score (total)	20.2 ± 3.9	15.3 ± 4.1	0.002
Sinus dominancy			0.012
Ethmoid > maxillary	8 (50%)	3 (21%)	
Ethmoid = maxillary	6 (40%)	4 (29%)	
Ethmoid < maxillary	2 (10%)	7 (50%)	
Olfactory function			
Total score	13.0 ± 6.2	17.2 ± 5.9	0.680
anosmia	13 (81%)	5 (36%)	0.028
hyposmia	2 (13%)	9 (64%)	
normosmia	1 (6%)	0 (0%)	
Blood eosinophil (%)	8.9 ± 5.2	4.0 ± 3.1	0.005
Atopic status	12 (75%)	10 (71%)	0.349
Bronchial asthma	8 (50%)	1 (7%)	0.001

ECRS: eosinophilic CRS; NECRS: non-eosinophilic CRS; CT: computed tomography.

## Data Availability

Data supporting this study can be obtained by contacting the corresponding author.

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
