# Peer review of "Aspergillus Enhances Eosinophil and Neutrophil Extracellular DNA Trap Formation in Chronic Rhinosinusitis"

_ijms, 2023, doi:10.3390/ijms242417264_

Round 1
Reviewer 1 Report
Comments and Suggestions for Authors
This study is in an attempt to investigate the effect of airborne fungi, Alternaria alternata and Aspergillus fumigatus, on eosinophil and neutrophil extracellular traps (EET and NET) formation. Nasal epithelial cells, eosinophils, and neutrophils were isolated from patients with eosinophilic chronic rhinosinusitis (ECRS), non-ECRS (NECRS) and healthy controls for the determination of eosinophil and neutrophil transepithelial migration, reactive oxygen species (ROS), fungal metabolic activity. ECRS was found to exhibit higher EET and NET formation, and peripheral blood eosinophils and neutrophils exhibit different immune responses against A. fumigatus according to the inflammatory status of the host. Results may therefore provide important insights into CRS pathogenesis.
Major comments
(1) The figure legend of Figure 2 in page 3 is wrongly labeled. Please check and correct. In this figure about fungal-induced eosinophil and neutrophil migration, the detailed information for the determination of the number and purity of eosinophils and neutrophils purified from patients’ peripheral blood should be provided as the circulating eosinophils should be very low in circulating blood and may be insufficient for the experiments. In this regard, what is the cell numbers of eosinophils and neutrophils used in the assay of eosinophil and neutrophil migration? In Figure 2(A), why there was no effect on eosinophils migration in NECRS upon Alt and Asp stimulation comparing with negative control? The full name of NIT in this Figure 2 should also be provided.
(2) In Figure 4 (A & B) of fluorescent staining, the PMA treated group should also be showed.
(3) In Figure 5, apart from IL-8, other CRS-related inflammatory cytokines such as IL-6, IL-1b, TNF-a should also be assayed. Instead of ELISA, ECP should be assayed by ImmunoCAP ECP fluorescent-enzyme immuno-assay (ThermoFisher Scientific).
(4) The detailed immunological and molecular mechanisms by which Aspergillus induced a larger amount of IL-8 production from neutrophils in ECRS and NECRS than Alternaria, and Aspergillus, but not Alternaria, induced more EET and NET formation in patients with ECRS compared to those with NECRS and healthy volunteers should be further discussed and illustrated.
(5) In Figure 7, besides ECRS, the results of the ROS and EET assay using eosinophils and neutrophils isolated from normal subjects and NECRS may also be included.
Minor comments
(1) N number should be provided in Figure 1.
(2) Since serum IgE concentrations are usually elevated in allergic fungal rhinosinusitis, it is suggested that the allergen specific IgE levels may also be included in Table 1.
Author Response
I thank the editors and referees of the ‘International Journal of Molecular Sciences’ for taking their time to review my article.
I have made some corrections in the manuscript after going over the referee’s comments.
Major comments
(1) The figure legend of Figure 2 in page 3 is wrongly labeled. Please check and correct.
Answer) Thank you for your accurate comment.
The entire contents of the Fig 2 legend have been rewritten.
In this figure about fungal-induced eosinophil and neutrophil migration, the detailed information for the determination of the number and purity of eosinophils and neutrophils purified from patients’ peripheral blood should be provided as the circulating eosinophils should be very low in circulating blood and may be insufficient for the experiments.
Answer)
Eosinophils were isolated from heparinized peripheral blood. With 40~50 mL of blood, we can isolate about 3~4 x 106 eosinophils.
Neutrophils were enough to use.
The purity and viability of cells were mentioned in section ‘4.3. Isolation of eosinophil and neutrophil’ as > 95%. (line 363 & 369)
In this regard, what is the cell numbers of eosinophils and neutrophils used in the assay of eosinophil and neutrophil migration?
Answer)
To clarify the number of cell applied on basolateral side of the transwell, ‘one hundred microliters of the eosinophils or neutrophils (5 × 106/mL)’ was added in section 4.5, line 403.
In Figure 2(A), why there was no effect on eosinophils migration in NECRS upon Alt and Asp stimulation comparing with negative control? The full name of NIT in this Figure 2 should also be provided.
Answer) Thank you for your comment.
Actually, we can not explain exactly the reason why eosinophil migration was not enhanced in NECRS. However, we tried to explain in Discussion, Line 244 to 246 as ‘These discrepancies may be associate with the strength of molecular interaction within the apical junctional complex or the difference in the effect of fungi on eosinophil-associated chemokine production from nasal epithelial cells.’
(2) In Figure 4 (A & B) of fluorescent staining, the PMA treated group should also be showed.
Answer) Figure 4 A & B were changed as recommended.
(3) In Figure 5, apart from IL-8, other CRS-related inflammatory cytokines such as IL-6, IL-1b, TNF-a should also be assayed. Instead of ELISA, ECP should be assayed by ImmunoCAP ECP fluorescent-enzyme immuno-assay (ThermoFisher Scientific).
Answer) Unfortunately we did not use ImmunoCAP for ECP.
In this study 2 × 105/ 200 μL of eosinophils and neutrophils were used for each experiments.
So the intracellular protein and chemical mediators level should be very low.
We measured IL-6 and TNF-α with ELISA kit.
IL-6 and TNF-α in NET were very low (0.1 ~ 0.4 pg/mL and 0.1 ~ 5.3 pg/mL). IL-6 and TNF-α in EET were about 0.1 ~ 1.4.0 pg/mL and 3.1~20.6 pg/mL.
So we added IL-6 and TNF-α in EET results as Figure S1.
And in Discussion Line 305 ‘The IL-6 and TNF-α levels in EETs were also higher in ECRS than in NECRS (Fig S2).’ was added.
(4) The detailed immunological and molecular mechanisms by which Aspergillus induced a larger amount of IL-8 production from neutrophils in ECRS and NECRS than Alternaria, and Aspergillus, but not Alternaria, induced more EET and NET formation in patients with ECRS compared to those with NECRS and healthy volunteers should be further discussed and illustrated.
Answer) Aspergillus strongly induced the development of EET and NET with the release of intracellular granules and chemical mediators.
To clarify, in Line 300-302, ‘Because Aspergillus strongly enhance the development of EET and NET than Alternaria, Aspergillus strongly induced the release of a larger amount of intracellular IL-8 from eosinophils and neutrophils.’ Was added.’ was added in Discussion.
And in Line 269-271, we briefly mentioned the reason why Alternaria did not influence EET development, as ‘Fungi exhibit remarkable diversity in terms of structure and components, and they inter-act with host and immune cell in various way. While Alternaria can activate eosinophils through various way, it lacks the ability to actively participate in the development of ETs as observed in Aspergillus.’
(5) In Figure 7, besides ECRS, the results of the ROS and EET assay using eosinophils and neutrophils isolated from normal subjects and NECRS may also be included.
Answer) We did same experiments with eosinophils and neutrophils isolated from NECRS patients and healthy volunteers. The results were very similar with ECRS.
To clarify Figure S1 was added.
And in Results, Line 199-201 and Line 208-210, ‘Although ROS production from eosinophils or neutrophils in NECRS patients was not so strong as ECRS, they showed similar pattern with ECRS (Fig S1 A and B).’ & ‘The effect of ROS inhibitors on EET and NET formation showed a similar pattern in cells isolated from NECRS and ECRS patients (Fig S1 C and D).’ were added.
Minor comments
(1) N number should be provided in Figure 1.
Answer) In Figure 1 legend, Line 82, ‘n=16 and 14’ was added
(2) Since serum IgE concentrations are usually elevated in allergic fungal rhinosinusitis, it is suggested that the allergen specific IgE levels may also be included in Table 1.
Answer) Thank you for your kind comment.
We decided atopic status of patients with skin prick test and MAST. We did not measure total IgE or allergen specific IgE at this time. So we cannot describe about that in Table 1.
I hope the revised manuscript will better meet the requirements of the ‘Pharmaceuticals’ for publication.
Thank you.
Reviewer 2 Report
Comments and Suggestions for Authors
Thank you very much for this interesting paper about the formation of EETs and NETs in patients with both eosinophilic and non-eosinophilic chronic rhinosinusitis (CRS). It seems that CRS patients may be more prone to fungal infection and/or hyperreactivity which is an important observation. The authors took under consideration two types of fungi, A. fumigatus and A. alternata. Could the authors comment briefly why one of them enhanced EET and NET formation and the other did not? Could the fact that half of the ECRS group had also asthma have any influence on the results? Would the results change if asthmatic patients would be excluded from the study? Please comment on these observations in brief as asthma is a risk factor in fungal infections. Could the authors add demographic data and the clinical profile of the control group?
Author Response
I thank the editors and referees of the ‘International Journal of Molecular Sciences’ for taking their time to review my article.
I have made some corrections in the manuscript after going over the referee’s comments.
The authors took under consideration two types of fungi, A. fumigatus and A. alternata. Could the authors comment briefly why one of them enhanced EET and NET formation and the other did not?
Answer) Although we cannot explain in detail, in Discussion Line 269-272 ‘Fungi have great diversity in terms of structure and components, and they interact with host and immune cell in various way. Although Alternaria can activate eosinophils in various way, they cannot actively participate in development of ETs as Aspergillus.’ Was added to explain about that.
Could the fact that half of the ECRS group had also asthma have any influence on the results? Would the results change if asthmatic patients would be excluded from the study? Please comment on these observations in brief as asthma is a risk factor in fungal infections.
Answer) Thank you for suggesting the important point.
When comparing the percentage of peripheral blood eosinophils in this study subjects, ECRS with asthma (10.2 ± 5.0 %) exhibited the highest percentage, followed by ECRS without asthma (7.6 ± 5.4 %) and NECRS (4.0±3.1 %).
Because, we did not specifically compare these three conditions, we cannot provide an exact answer to you point.
To address the potential impact of asthma comorbidity on the formation of EET and NET, in Line 274~277, ‘In this study, half of ECRS patients had asthma. Although, we did not directly compare the impact of asthma comorbidity on EET formation, the biological properties of eosinophils in asthmatic patients may predispose to the development of EET.’ was added in Discussion .
Could the authors add demographic data and the clinical profile of the control group?
Answer) For the isolation of neutrophils and eosinophils, the control group comprised healthy volunteers, and demographic data and the clinical profile were not specified.
Regarding nasal epithelial cell culture, only the mean age was mentioned in Line 375.
I hope the revised manuscript will better meet the requirements of the ‘Pharmaceuticals’ for publication.
Thank you.
Round 2
Reviewer 1 Report
Comments and Suggestions for Authors
The manuscript has been revised properly according to reviewer's comments, questions and suggestions.
Author Response
Thank you for your kindness.